

# Effect of dynamic taping on neck pain, disability, and quality of life in patients with chronic non-specific neck pain: a randomized sham-control trial

Mohammad Sidiq[1], Aksh Chahal[1], Balamurugan Janakiraman[2,3], Faizan Kashoo[4], Sharad Kumar Kedia[5], Neha Kashyap[6], Richa Hirendra Rai[7], Neha Vyas[8], T.S. Veeragoudhaman[3], Krishna Reddy Vajrala[1], Megha Yadav[1], Shahiduz Zafar[1], Sanghamitra Jena[1], Monika Sharma[1], Shashank Baranwal[9], Mshari Alghadier[10], Abdullah Alhusayni[11], Abdullah Alzahrani[11] and Vijay Selvan Natarajan[12]

[1] Department of Physiotherapy, School of Allied Health Sciences, Galgotias University, Greater Noida, Uttar Pradesh, India
[2] Department of Physiotherapy, School of Allied Health Sciences, Madhav University, Abu Road, Sirohi, Rajasthan, India
[3] SRM College of Physiotherapy, Faculty of Medicine and Health Sciences, SRM Institute of Science and Technology (SRMIST), Kattankulathur, Tamil Nadu, India
[4] Department of Physical Therapy and Health Rehabilitation, College of Applied Medical Sciences, Majmaah University, Majmaah, Riyadh, Saudi Arabia
[5] Department of Physical Medicine & Rehabilitation, NIMS University Hospital, Jaipur, Rajasthan, India
[6] Physiotherapy, Maharishi Markandeshwar Deemed to Be University, Ambala, Haryana, India
[7] Physiotherapy, Delhi Pharmaceutical Sciences and Research University, New Delhi, Delhi, India
[8] Physiotherapy, University of Engineering and Management, Jaipur, Rajasthan, India
[9] Nims College of Physiotherapy and Occupational Therapy, NIMS University, Jaipur, Rajasthan, India
[10] Department of Health and Rehabilitation Sciences, College of Applied Medical Sciences, Prince Sattam bin Abdulaziz University, Alkharj, AR Riyadh Province, Saudi Arabia
[11] Department of Rehabilitation Sciences, College of Applied Medical Sciences, Shaqra University, Shaqra, West Province, Saudi Arabia
[12] Physiotherapy, KMCT College of Allied Medical Sciences, Manassery, Kozhikode, Kerala, India

Corresponding author
Mshari Alghadier,
m.alghadier@psau.edu.sa

## ABSTRACT

**Background**. In 2020, 203 million people experienced neck pain, with a higher prevalence in women. By 2050, it is predicted that neck pain will affect 269 million people, representing a 32.5% increase. Physical rehabilitation is often employed for the treatment of chronic non-specific neck pain (CNSNP) and the associated functional loss. Taping is frequently used as an adjunct treatment alongside primary physical rehabilitation. Unlike kinesio tape (KT), the therapeutic benefits of dynamic tape (DT) have not been thoroughly explored and documented in non-athletic conditions. Therefore, the aim of this trial was to determine the effects of DT on pain, disability, and overall well-being in individuals experiencing CNSNP.

**Methods**. A prospective parallel-group active controlled trial was conducted at a single center, involving 136 patients with CNSNP, randomly allocated in a 1:1 ratio. The sham taping group (STC) received standard physiotherapy care ($n = 67$) alongside DT without tension, while the dynamic taping group (DTC) ($n = 69$) underwent standard cervical offloading technique with appropriate tension in addition to standard

physiotherapy care. Demographic information and three patient-reported outcome measures (PROMs), namely the Neck Disability Index (NDI), Visual Analogue Scale (VAS), and the World Health Organization—Five Well-Being Index (WHO-5), were collected for each participant at three time points (baseline, four weeks post-taping, and four weeks follow-up).

**Results**. At baseline, no significant differences were observed between the STC and DTC for any outcome measure. Notably, all three PROMs exhibited a significant improvement from baseline to four weeks post-intervention, with moderate to small effect sizes (NDI $\eta p^2 = 0.21$, VAS $\eta p^2 = 0.23$, and WHO-55 $\eta p^2 = 0.05$). The WHO-5 scores for both groups demonstrated improvement from baseline through follow-up ($p < 0.001$). The NDI and VAS scores ameliorated from baseline to the four weeks post-taping period, with marginal improvements observed during the four weeks follow-up.

**Conclusion**. The incorporation of DT as an adjunct to standard physiotherapy care yielded enhancements in pain levels, functional disability, and well-being among patients with CNSNP when compared to the sham group. However, the sustainability of these improvements beyond the taping period lacks statistical significance and warrants further validation.

fic neck pain, Disability, Quality of life, Sham taping, Cervical offload technique

## BACKGROUND

Global Burden of Disease (GBD) 2019 study revealed that since 2010, neck pain consistently ranks among the top five causes of global disability-adjusted life years in individuals aged 25 to 74 years (*Safiri et al., 2020*). Neck pain management presents a complex array of challenges, encompassing both medical and lifestyle factors. A primary challenge lies in precisely diagnosing the underlying causes of neck pain, which may originate from diverse sources such as muscle strain, poor posture, herniated discs, or even stress (*Hogg-Johnson et al., 2009*). This diversity complicates tailored treatment, often necessitating a combination of medical interventions, physical therapy, and lifestyle adjustments (*Gross et al., 2010*). Furthermore, the subjective nature of pain perception introduces an additional layer of complexity, rendering it challenging to quantify and monitor improvements (*Melzack & Wall, 1994*). As the prevalence of neck pain continues to rise, attributed to increasingly sedentary and technology-centric lifestyles (*Sarig-Bahat, 2003*), addressing these multifaceted challenges becomes crucial for enhancing the quality of life among individuals with neck pain (*Cagnie et al., 2007*). Incorporating holistic approaches that address not just physical symptoms but also psychological and ergonomic contributors is essential (*Meade et al., 2019*). Taping emerges as a promising adjunct in the comprehensive management of multifaceted conditions such as chronic non-specific neck pain (CNSNP). Integrating taping into the therapeutic arsenal not only diversifies treatment options but also highlights the potential efficacy of this modality. Notably, the specific impact of novel taping methods remains a relatively unexplored avenue within the field of neck pain

management, warranting further investigation and scrutiny to elucidate its effects and contribute to the evolving landscape of evidence-based interventions. Hence, adequately powered studies are needed to generate sufficient evidence supporting the use and clinical efficacy of taping (*Zhang et al., 2016*).

The physical and mechanical properties of tapes used for therapeutic purpose and to improve performance in athletes have evolved since 1986 (*McConnell, 1986*). Recent tape designs aim to reduce the risk of skin reactions, allergies, skin traction injuries, movement restrictions, tape fatigue, and unnatural movement patterns, leading to frequent clinical application and considering it a valuable adjunct in musculoskeletal rehabilitation (*Taylor, O'Brien & Brown, 2014*). Simultaneously, the lack of consensus on the biomechanical principles at work, acceptable taping techniques, quantity of tension during application, and application frequency with dosage are still debatable. Moreover, the utility of kinesio-tape (KT) to treat musculoskeletal conditions in athletic population and other musculoskeletal conditions is well documented (*Mostafavifar, Wertz & Borchers, 2012*; *Ferreira, Resende & Roriz, 2017*; *Tudini et al., 2023*). The introduction of dynamic tapes (DT), characterized by distinct mechanical properties compared to traditional KT, represents a potential advancement in therapeutic applications. Notably, these DT exhibit superior attributes such as increased elasticity, facilitating unrestricted movements post-taping, four-directional breathability, and enhanced tissue offloading capabilities when compared to conventional KT. As a variant of mechanical tapes, DT promise to provide adequate tissue offloading for areas requiring rest or compromised function without interfering with movement patterns and overall function.

However, there is a lack of clarity in the evidences regarding the effect of DT, including its application site, frequency, duration, and impact on pain due to tissue off-loading, as well as its retention effects. Therefore, the purpose of this study was to determine the efficacy of adding the DT technique to standard physiotherapy care. The primary objective of this research is to investigate the efficacy of DT method in alleviating pain, reducing functional disability, and enhancing mental well-being among patients with CNSNP.

## METHODS

### Study design
A parallel-arm randomized controlled trial was prospectively registered with the Clinical Trial Registry India under the identifier 2022/07/043700. The trial aimed to compare the effects of DT against sham taping, conducted in accordance with the Consort—2010 guidelines (*Calvert et al., 2013*). All participants provided informed consent, and the trial adhered to the principles outlined in the Declaration of Helsinki (*Skierka & Michels, 2018*). Ethical approval was obtained from the Departmental Ethics committee of the Physiotherapy Department, NIMS University, affirming adherence to ethical standards and oversight for the research NU/NCPT/JUNE/15. Based on the criteria for inclusion, an independent observer used block procedure with a block size of 6 for the randomization of participants in a 1:1 allocation ratio into the DT or sham group (*Kang, Ragan & Park, 2008*). Index cards with sequential number were folded and sealed in opaque envelopes.

The primary investigator was concealed from this task. The assessors were blinded to the opened envelop and group assignment. This study adhered to the Sex and Gender Equity in Research (SAGER) guidelines for reporting sex and gender information.

## Participants and recruitment

Participants with a history of CNSNP were recruited over a 1-year period from the Physical and Rehabilitation Medicine (PMR) department and the outpatient department (OPD) of physiotherapy at the National Institute of Medical Sciences (NIMS) University teaching hospital. Additionally, a targeted advertising campaign, utilizing pamphlets, was implemented to raise awareness about the research within the nearby residential zones, covering an approximate area of 3 square kilometers surrounding the study center. This initiative aimed to enhance recruitment efforts and engage potential participants in the study. NIMS hospital, situated in the densely populated Jaipur city, India, provides free consultation, diagnostics, and treatments, making it a major tertiary care center in the region. Two trained physiotherapists screened all potential participants, and eligible individuals who provided written consent to participate were recruited. Data collection from participants commenced in July and concluded in December 2022.

The inclusion criteria encompassed individuals aged 18 to 60 years with a duration of CNSNP lasting at least three months. Diagnosis was established through a comprehensive assessment of medical history, physical examination, and the exclusion of specific underlying pathologies *via* imaging studies and laboratory tests. Criteria for inclusion further specified chronic neck pain without a clear structural cause, the absence of identifiable systemic or local pathology, and a minimum duration of three months. Exclusion criteria comprised patients diagnosed with fibromyalgia, those who had undergone taping in the cervical region in the past 6 months, individuals with whiplash injuries, tape-related allergies, cervical myelopathy, a history of cervical and/or thoracic spine fracture, inflammatory conditions, skin conditions involving the cervical region, and pregnant women (see Fig. 1).

## Sample size determination

The G*power software, version 3.1.9.4 (University of Kiel, Kiel, Germany) was used to determine the required power-calculated sample for this two tailed trial (*Faul et al., 2007*). A two-group independent *t*-test with a significance level of 0.05 and a power of 0.80 was employed. The analysis resulted in an optimal sample size of 140 participants, evenly distributed with 70 in each group (*Naseri et al., 2016*).

## Outcome measures

The primary outcomes of interest included functional disability, measured using the Hindi version of the neck disability index (NDI), and the intensity of neck pain during rest using the visual analogue scale (VAS). Additionally, the quality of life was assessed with the World Health Organization—Five Well-Being Index (WHO-5). The testing of outcome measures was conducted individually in a quiet ambience at the physiotherapy outpatient department (OPD) and lasted for 15 to 20 min. The outcome measures were recorded

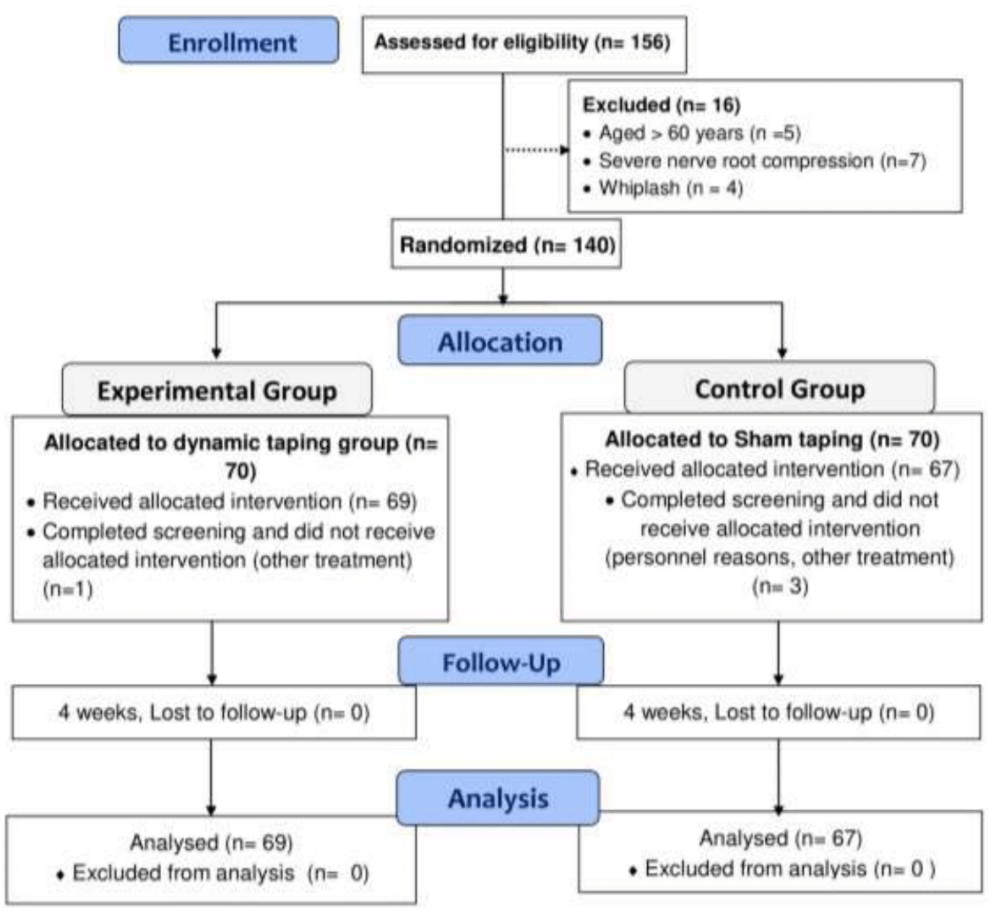

**Figure 1** Comprehensive study flow chart: recruitment, randomization, and follow-up.

at baseline (pre-taping), post-intervention (at 4 weeks post-taping), and follow-up (at 4 weeks after the last taping session).

### Neck disability index

The functional disability related to neck pain was assessed using the NDI, a well-established, commonly used, reliable, and valid patient-reported tool (*MacDermid et al., 2009*; *Jørgensen et al., 2014*; *Young et al., 2019*). The NDI comprises 10 items, with six possible exclusive responses for each item, and the total possible scores range from '0' to '50'. A higher NDI score indicates more severe disability, and the NDI scores can be expressed as a disability percentage by doubling the obtained scores (*MacDermid et al., 2009*).

### Visual analogue scale

The intensity of neck pain during rest was determined using VAS. A horizontal line of 10 cm in length was utilized, where 0 cm (beginning of the line) represents 'no pain', and 10 cm represents 'worst pain' (the end of the line). VAS is a self-reported assessment of pain intensity, recognized as a valid, reliable, and widely used outcome measure in clinical practice (*Delgado et al., 2018*).

### World Health Organization—Five Well-Being Index

The WHO-5 is a short-form patient-reported measure of mental well-being commonly utilized in clinical practice and research (*Topp et al., 2015*). The WHO-5 comprises five statements with a Likert scale ranging from '0' to '5', and the total score ranges from 0 to 25. This total score is then multiplied by 4 to derive the percentage score. A percentage of 0% represents "worst well-being", while 100% signifies "best well-being" (*Topp et al., 2015*).

### Physical activity interview

In this study, two physical therapists, actively engaged in both pre and post assessments of participants, conducted face-to-face interviews to measure participants' levels of physical activity. The assessment of physical activity levels in this study employed a categorization system derived from a recent research article (*Sidiq et al., 2021*). Participants were categorized into mild physical activity if they reported no engagement in any form of physical activity in the previous week. Moderate physical activity was assigned to individuals who participated in any form of aerobic training such as swimming, bicycling, or running 2–3 times in the last week. Vigorous physical activity was designated for those involved in aerobic training for more than 3 days or engaged in weight lifting 1–3 days in the last week.

## Description of intervention (experimental and standard physiotherapy care protocol)

Six qualified and experienced physical therapists affiliated with NIMS participated in this study. Among them, two therapists (blinded to subject allocation) conducted baseline, post-test, and follow-up outcome measurements. Another two physical therapists (blinded to subject allocation) provided standard physiotherapy care. The remaining two therapists underwent a 3-day training conducted by the principal author (MS) before the study commenced. The principal author (MS) played a crucial role in maintaining consistency throughout the study, ensuring uniformity in standard physiotherapy care, taping application, and pre-post assessments across all six physical therapists. Typically, the taping session lasted 10–15 min, while standard physiotherapy care required 45 min.

Each week, 24–28 patients were managed simultaneously, distributed among six physical therapists. The schedule involved treating 6-7 patients (3-days a week) on Monday, Wednesday, and Saturday, and an additional 6–7 patients on Tuesday, Thursday, and Friday. This arrangement totaled 12–14 patients for each therapist individually and 24–28 for the two therapists working collaboratively. The data collection spanned six months, during which the center successfully collected data for 136 patients with CNSNP, despite encountering 4 dropouts during the study period.

### DTC group protocol (experimental group)

After screening for recruitment criteria, a sensitivity test was conducted by applying a small piece of DT to the inner part of the non-dominant arm of the volunteer participant, kept in place for 24 h. The following day, the test area was examined for any allergic reaction. The trial utilized a waterproof tape with a porous texture, adhesive, and a width of five cm, manufactured by Posture Pals Pvt. LTD, Port Vila, Vanuatu.

Before tape application, the treatment area was cleaned with water, and excess hair was trimmed using a trimmer. The skin was kept dry and free from any oils, lotions, or sweat for proper adhesion. The tape was cut to the desired length, rounding the edges to prevent peeling. Patients were informed about the features of DT, such as its four-way stretch materials and strong recoil property. The starting position involved both hands at the side of the body. Patients were instructed to shrug both shoulders while keeping the scapulae in a retracted position. The tape was then applied, running through both shoulders using the cervico-thoracic junction while maintaining C7 as the fulcrum point. Anchor points were created at both ends of the tape with no stretch applied, applying light tension to the skin. Patients were then instructed to shrug their shoulders while retracting both scapulae, and the tape was applied using 20–30% stretch or tension. It was smoothed down from the cervico-thoracic junction to the other side of the shoulder and anchored without any tension. Patients were asked to relax their shoulders, ensuring they felt the weight of the shoulders borne by the tape. The last component of the cervical offload technique involved the box offload technique through the supraclavicular area. Two small strips were cut for both sides of the neck. The anchor points were created around the spine of the scapula, pulling the soft tissue with mild tension, and then stuck around another anchor point anteriorly just below the clavicle, as seen in Fig. 2. Participants were blinded to appropriate tension taping and (sham) no tension taping intervention. The cervical offload technique was employed during the application of dynamic tape. Standard physiotherapy care for both groups was similar, with only the adjunct taping intervention being manipulated. The tape was applied and replaced every 3 days for the next 4 weeks, and the skin under the tape was examined during reapplications. Both groups received nine sessions of taping, and the ninth dynamic tape application was removed on the last day of the 4 weeks' post-intervention period (see Fig. 2).

### STC group protocol (sham group)

The standard physiotherapy care protocol was consistent across all participants, ensuring uniform intensity, duration, and sequence for each component. Therapeutic exercises focus on strengthening the neck, shoulder, and upper back muscles, along with stretching exercises to enhance flexibility. Transcutaneous electrical nerve stimulation was employed for pain management. Additionally, participants received education on posture and ergonomics, cervical traction, activity modification, and a prescribed home exercise program to maintain progress. Further details regarding standard physiotherapy care are available in File S1.

## Statistical analysis

IBM SPSS Statistics for Windows, Version 20.0 (IBM Corp., Armonk, NY, USA) was utilized for the analysis. Baseline measures and socio-demographic clinical characteristics were analyzed using chi-square test, ANOVA, and independent sample $t$-test. The scores obtained from the NDI, VAS, and the WHO-5 were approximately normally distributed for both the experimental and sham groups, as indicated by the Shapiro–Wilk test ($p > 0.05$) and a visual examination of their histograms, normal Q–Q plots, and box plots. A 2

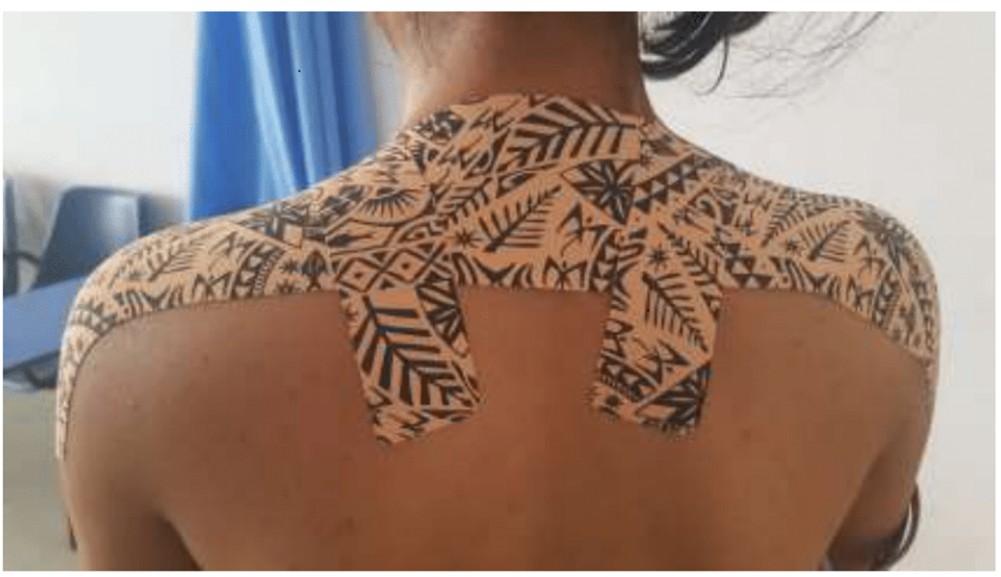

**Figure 2  Tape application protocol for the dynamic tape group.**

(condition) ×3 (time) mixed model repeated measures ANOVA, with group (dynamic taping and sham taping), and measurement times (baseline, post-intervention, and 4 weeks follow-up) as between and repeated measures respectively, was conducted. Levene's and Mauchly's tests were used to check the assumptions of homogeneity and sphericity respectively. In the case of sphericity assumption violations, the Greenhouse-correction method was applied (*Mishra et al., 2019*). The Bonferroni analysis was conducted to determine the difference within and between groups (*Kim, 2015*; *Haverkamp & Beauducel, 2017*). In addition, the effect size was calculated as partial eta squared ($\eta p^2$), and the level of significance for the statistical tests was set at $p < 0.05$ with a 95% confidence interval. The participants completed the questionnaire outcome measures (NDI, VAS, and WHO-5) at baseline, 4 weeks post-intervention, and 4 weeks follow-up. The improvement or changes in scores from baseline to follow-up were expressed as a percentage by calculating the mean difference between the baseline and follow-up scores to the ratio of the baseline score expressed as a percent. The hypothesis of interest was the group*time interaction at a priori alpha ($\alpha$) level of 0.05. Supplementary data analyses were performed according to the intention-to-treat principle to preserve the original random assignment, using the first observation carried forward method, assuming that the pre-taping data is most representative (important moment of effect) of post-intervention measurements for drop-outs (File S2). According to the recommendation of SAGER guidelines, gender difference and/or similarities were analyzed.

## RESULTS

### Participants

Table 1 presents the descriptive statistics of the participants who completed the baseline questionnaire in this trial. Initially, 140 patients with CNSNP volunteered to participate. After 1:1 allocation and baseline assessment, four patients dropped out (DTC group $n = 1$ and STC group $n = 3$) after one taping session. The flow chart (Fig. 1) describes the reasons for dropout and exclusion ($n = 16$) from the study. The participants had a mean age of $43.76 \pm 8.15$, with a majority being female ($n = 84$) 61.8%, DTC group ($n = 46$) 66.7% *versus* STC group ($n = 38$) 56.7%, $p < 0.01$ as shown in Table 1. The Chi-square test indicated significant differences between the groups at baseline for variables such as gender, type of occupation, self-reported level of physical activity, and smoking habits . However, the independent sample t-tests revealed no differences between the groups on any of the baseline outcome measures. Adverse effects or events reported during the taping sessions included a feeling of restriction in neck movements ($n = 2$), redness of the skin under the tape ($n = 3$), which resolved in 2 days. Importantly, none of the adverse events resulted in the discontinuation of the intervention.

### Effect of treatment

To assess the impact of the intervention on functional disability related to neck pain, a 2 (intervention groups) × 3 (time points) repeated measures ANOVA was conducted. Differences between STC and DTC were explored using Bonferroni-corrected post hoc tests to determine the variations between taping interventions. The results of the NDI demonstrated a significant main effect for time ($F = 219.24$, $p < 0.001$, $\eta p^2 = 0.62$). Although the interaction effect was significant ($F = 35.67$, $p < 0.001$, $\eta p^2 = 0.21$), the effect size considerably reduced, indicating a smaller effect of NDI scores between groups. The mean NDI scores in percentage over time for both the groups are presented in Fig. 3. The post-hoc analysis (Bonferroni correction) revealed significant differences between baseline NDI scores and post-intervention NDI scores (mean difference = 10.24 (8.81–11.67), $p < 0.001$). However, the NDI scores between post-intervention and the follow-up period were not significant (mean difference = 0.285 (−0.79–1.36), $p = 0.91$), suggesting that the improvement in neck function occurred until the taping intervention but did not improve further. Within the DTC group, a significant reduction in functional disability scores (NDI) was observed between baseline and post-intervention (mean difference = −13.76, $t = 14.3$, $p < 0.001$), but not between post-intervention and follow-up (mean difference = 1.4, $t = 2.46$, $p = 0.16$). Standard physiotherapy care with sham taping resulted in a 13.4% improvement compared to DT with tension, which resulted in a 35.4% improvement (Table 2).

A repeated measures ANOVA was conducted for pain intensity (VAS) and WHO-5, for each measures 2 (intervention group) × 3 (time points) was tested with the Bonferroni post-hoc tests. The 2 × 3 mixed ANOVA of VAS score demonstrated a significant main effect of intervention for time ($F = 305.2$, $\eta p^2 = 0.69$, $p < 0.001$) with a large effect size, and a significant interaction between intervention and time ($F = 26.6$, $\eta p^2 = 0.166$, $p < 0.001$) with a small effect size. Bonferroni comparison analysis indicated

**Table 1 Demographic data.**

| Variables | All samples (n = 136) | DTC (n = 69) | STC (n = 67) | p value |
|---|---|---|---|---|
| Age (years, mean ± SD) | 43.76 ± 8.15 | 42.74 ± 8.7 | 44.82 ± 7.43 | 0.137 |
| [a]BMI kg/m$^2$ | 25.7 ± 3.1 | 25.97 ± 2.97 | 25.52 ± 3.24 | 0.40 |
| Chronicity of neck pain (Mo) | 44.3 ± 24.1 | 45.31 ± 19.37 | 43.4 ± 28.24 | 0.64 |
| [a]Gender, n (%) | | | | |
| Male | 52 (38.2) | 23 (33.3) | 29 (43.3) | **0.01** |
| Female | 84 (61.8) | 46 (66.7) | 38 (56.7) | |
| [a]Marital status, n (%) | | | | |
| Married | 75 (55.1) | 35 (50.7) | 40 (59.7) | 0.19 |
| Not married/single | 61 (44.9) | 34 (49.3) | 27 (40.3) | |
| [a]Occupation, n (%) | | | | |
| Clerical/home maker | 38 (27.9) | 14 (20.3) | 24 (35.8) | **0.009** |
| Business | 26 (19.1) | 20 (29) | 06 (9) | |
| Professionals | 24 (17.6) | 14 (20.3) | 10 (14.9) | |
| Skilled labor | 48 (35.3) | 21 (30.4) | 27 (40.3) | |
| [a]Level of activity n (%) | | | | |
| Low | 27 (19.9) | 08 (11.6) | 19 (28.4) | **0.044** |
| Moderate | 80 (58.8) | 48 (69.6) | 32 (47.8) | |
| Vigorous | 29 (21.3) | 13 (18.8) | 16 (23.9) | |
| [a]Smoking, n (%) | | | | |
| Yes | 30 (22.1) | 07 (10.1) | 23 (34.3) | **0.001** |
| No | 106 (77.9) | 62 (89.9) | 44 (65.7) | |
| [b]NDI, mean ± SD | 44.05 ± 9.5 | 42.89 ± 4.52 | 44.1 ± 9.55 | 0.364 |
| VAS (0–10) | 6.25 ± 1.38 | 6.22 ± 1.43 | 6.31 ± 1.34 | 0.73 |
| WHO 5 Index | 51.5 ± 13.1 | 53.4 ± 14.48 | 49.59 ± 11.26 | 0.09 |

**Notes.**

The values are presented as proportion and percentage (%) for categorical variable, indicated by [a]Chi-square.

Student $t$ test was used for continuous variable and expressed as mean and standard deviation.

SD, Standard deviation; DTC, Dynamic taping with conventional physiotherapy; STC, Sham taping with conventional physiotherapy; Mo, months.

[b]NDI expressed as 100 percent by doubling the score.

Bold values indicate statistical significance.

significant differences between baseline pain and post-intervention pain VAS scores (mean difference = 1.99 [1.76–2.23], $p < 0.001$), as well as post-intervention pain and follow-up pain scores (mean difference = 0.377 [0.17–0.59], $p < 0.001$) (see Fig. 4). The DTC and the STC exhibited a substantial difference in the percentage of improvement between baseline and follow-up measures (47.26% *versus* 28.4%). The mental wellbeing measures (WHO-5) showed a significant main effect of intervention ($F = 7.32$, $\eta p^2 = 0.05$, $p = 0.008$) but did not demonstrate significant interaction between intervention and time effect ($F = 3.38$, $\eta p^2 = 0.0525$, $p = 0.078$). Further, both groups showed a nearly similar improvement percentage in WHO-5 scores between baseline and follow up (23.4% *versus* 19.9% respectively). A supplementary analysis for gender difference for the main effect and interaction effect (time*gender) showed no significant differences for the outcome

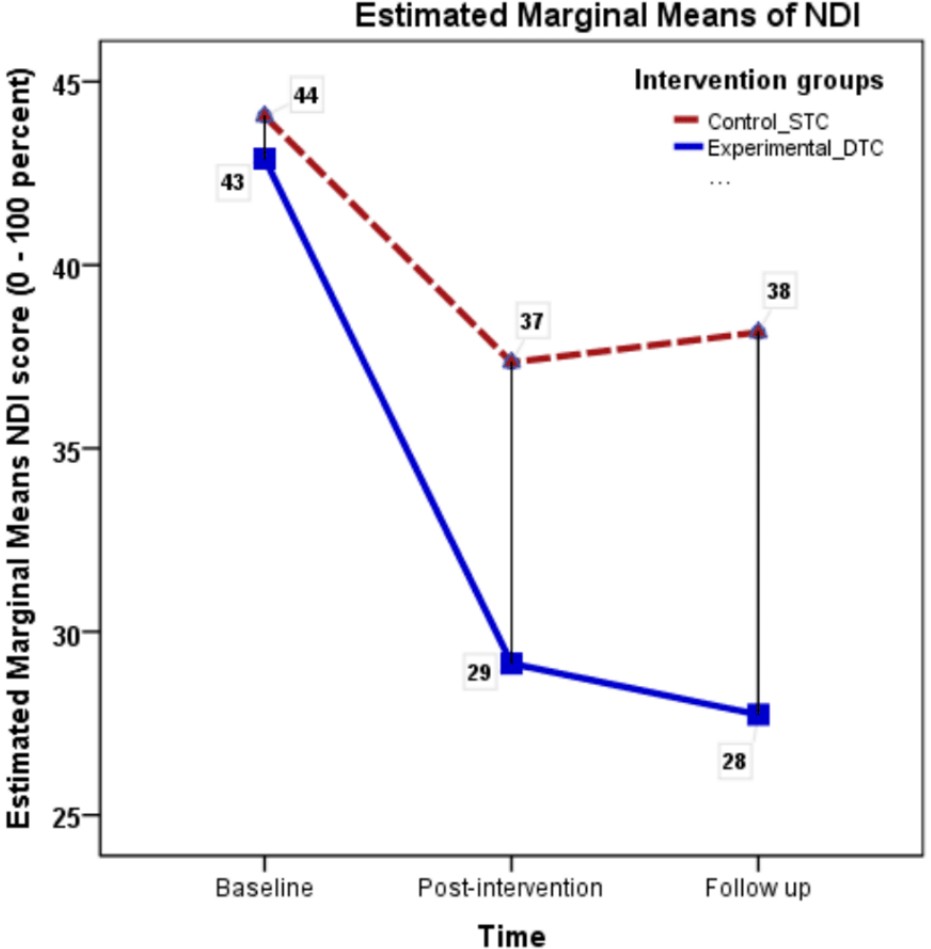

**Figure 3** Neck Disability Index (NDI) mean scores across measurement occasions under two conditions.

measures NDI, VAS, and WHO-55 ($F = 3.13$, $p = 0.80$, $\eta p^2 = 0.023$, $F = 0.339$, $p = 0.67$, $\eta p^2 = 0.03$, $F = 0.64$, $p = 0.46$, $\eta p^2 = 0.04$ respectively) (see Fig. 5).

## DISCUSSION

The findings of this two-arm parallel trial, including patients with CNSNP in both experimental and sham groups, revealed significant improvements in pain (VAS), functional disability (NDI), and mental wellbeing (WHO-5). However, the DTC demonstrated greater improvements in all three outcome measures. The findings of this study provide evidence supporting the short-term effectiveness (4 weeks) of clinically applying DT as an adjunct to standard physiotherapy care for patients with NSCNP. The observed superior improvement in the DTC group may attributed to factors such as enhanced position sense, tissue offloading, reduced muscle fatigue, controlled deceleration loading, facilitated target muscle engagement, and increased afferent input to the central

**Table 2  Results of comparison the outcome measures within the groups and between the placebo (STC, $n = 67$) group and experimental (DTC, $n = 69$) groups.**

| Variables | DTC | STC | MD (95% CI) | $\eta p2$ | P |
|---|---|---|---|---|---|
| Functional disability (NDI) | | | | 0.86 | |
| Baseline | 42.9 ± 4.5 | 44.05 ± 9.5 | 1.16 (−1.36, 3.68) | | 0.36 |
| Post-intervention | 29.13 ± 9.1 | 37.34 ± 9.8 | 8.21 (5.32, 11.1) | | 0.01 |
| Follow-up | 27.7 ± 6.7 | 38.16 ± 9.2 | 10.42 (7.7, 13.1) | | 0.001 |
| [a]Improvement (%) | 35.4% | 13.4% | | | |
| Greenhouse-Geisser | 193.4 | 46.19 | | | |
| p value | 0.001 | 0.001 | | | |
| Pain intensity (VAS) | | | | 1.3 | |
| Baseline | 6.22 ± 1.43 | 6.29 ± 1.34 | 0.08 (−0.39, 0.56) | | 0.7 |
| Post-intervention | 3.51 ± 0.82 | 5.01 ± 1.12 | 1.51 (1.17, 1.83) | | 0.001 |
| Follow-up | 3.28 ± 0.95 | 4.5 ± 1.35 | 1.22 (1.1, 1.59) | | 0.02 |
| [a]Improvement (%) | 47.26% | 28.4% | | | |
| Greenhouse-Geisser | 221.37 | 94.3 | | | |
| p value | 0.001 | 0.001 | | | |
| WHO-5 Index | | | | 0.48 | |
| Baseline | 53.4 ± 14.4 | 49.6 ± 11.3 | 3.8 (−8.2, 0.58) | | 0.09 |
| Post-intervention | 57.6 ± 14.2 | 51.64 ± 10.6 | 5.9 (1.71, 10.2) | | 0.006 |
| Follow-up | 65.8 ± 12.9 | 59.5 ± 8.7 | 6.3 (2.6, 10.13) | | 0.001 |
| [a]Improvement (%) | 23.2% | 19.9% | | | |
| Greenhouse-Geisser | 172.48 | 64.5 | | | |
| p value | 0.001 | 0.001 | | | |

**Notes.**

The values are presented as proportion and percentage (%) for categorical variable, indicated by [a]Chi-square.

Student $t$ test was used for continuous variable and expressed as mean and standard deviation.

SD, Standard deviation; DTC, Dynamic taping with conventional physiotherapy; STC, Sham taping with conventional physiotherapy; Mo, months.

nervous system (*Kashoo & Ahmad, 2020*). However, caution must be exercised when interpreting the results of this study due to the lack of control over potential confounding factors, such as variations in tape application between therapists and the subjective nature of the instruments used. Doubts regarding compliance with the home program may also have influenced the outcomes. Moreover, the trial design did not permit an exclusive estimation of the possible placebo effect of DT on self-reported outcome measures. Thus, it remains challenging to discern whether the effects observed in the sham group are due to the placebo tape effect, standard physiotherapy care, or natural changes over time. Addressing this uncertainty is particularly relevant, as a systematic review of 60 clinical trials on neck pain suggested that a 38% reduction in pain scores could be attributed to the placebo effect (*Hu et al., 2023*). Despite these challenges, efforts were made to mitigate the impact of confounding factors by monitoring and ensuring procedural consistency throughout the study.

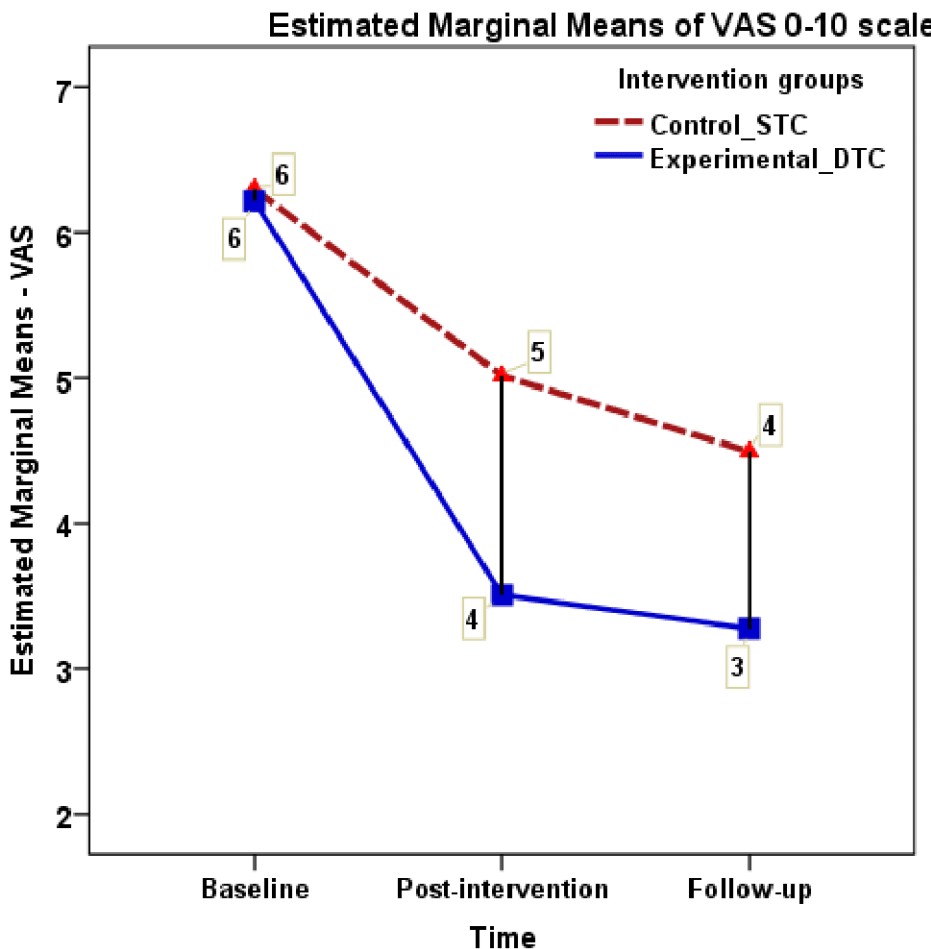

**Figure 4** Visual analogue scale mean scores across measurement occasions under two conditions.

## Effect of DT on pain (VAS)

Our study revealed a statistically significant decrease in VAS scores from baseline to follow-up, demonstrating a reduction of 47.26% and 28.4% in the DTC and STC groups, respectively. Notably, both groups surpassed the Minimal Clinically Important Difference (MCID) for VAS, which is reported for CNSNP at −21% (*Lauche et al., 2013*). Consistent with our findings, a clinical trial comparing KT to DT in 30 patients with non-specific low back pain reported a 47.9% improvement in VAS scores for the DT group (Mean = 5.80 to 3.71), while the KT group exhibited a 28.0% improvement (Mean = 5.7 to 4.5) from baseline to the 3rd day post-treatment (*Jain, 2022*). Similarly, a study conducted in Korea involving 40 participants with chronic neck pain for 4 weeks (three times a week) reported a 50% reduction in VAS scores compared to 46.5% in the sham group (*Yoon & Kim, 2022*).

## Effect on disability (NDI)

Our study reported a 13.7% reduction in neck disability from baseline to post-test scores, indicating a total improvement of 37%, while studies suggest that the MCID for NDI is

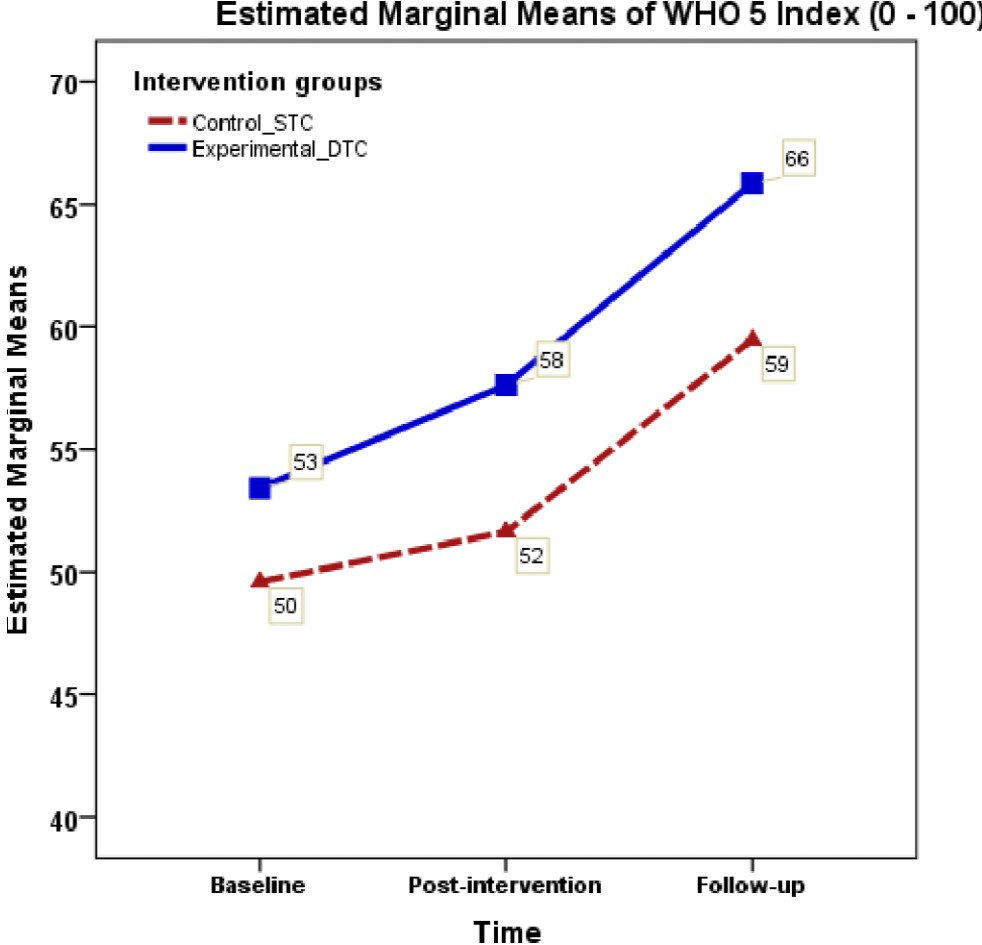

**Figure 5** WHO-5 index mean scores across measurement occasions under two conditions.

approximately −10% (*Lauche et al., 2013*). Similarly, a study reported a 45.6% reduction in disability after 4 weeks (three times a week) in the DT group and 34.6% in the Sham group (*Yoon & Kim, 2022*). These changes in NDI scores may be attributed to improved cervical muscle activity, potentially leading to a further reduction in muscle fatigue (*Zulfikri & Justine, 2017*), and the superior mechanical properties of DT fostering favorable postural habits among participants with CNSNP (*McNeill & Pedersen, 2016*). Additionally, research indicates that DT outperforms KT in enhancing muscle endurance (*Alahmari et al., 2020*). Remarkably, our study also demonstrated improvement in the sham group without tension. These results align with research suggesting that taping without tension can provide additional afferent inputs and enhance positional sense (*Alahmari et al., 2017*; *Reddy, Maiya & Rao, 2012*; *Kilinç, Harput & Baltaci, 2015*; *Vanti et al., 2015*; *Öztürk et al., 2016*; *Alahmari et al., 2017*; *Ay et al., 2017*). However, contrasting were reported by *De Jesus et al. (2017)* and *Pinheiro et al. (2021)*, indicating no change in muscle function with different tape tensions.

A comprehensive review of studies conducted on DT has revealed heterogeneous outcomes. For instance, *Esen et al. (2022)* reported a significant reduction in navicular drop distance following DT application in adolescent female volleyball players. Conversely, *Rengaramanujam (2021)* found no significant immediate or short-term differences between DT and KT in individuals with chronic non-specific low back pain regarding pain, disability, mobility, and kinesiophobia. DT has proven to be effective in addressing other musculoskeletal conditions, as evidenced by a study that investigated its role in managing greater trochanteric pain syndrome (GTPS). The study analyzed the impact of DT on specific gait-related parameters and pain reduction (*Robinson et al., 2019*). Fifty women with GTPS underwent three-dimensional gait analysis with active and sham tape applications. The results indicated a significant reduction in hip adduction moment, adduction angle, internal rotation, and pelvic obliquity, with meaningful pain reduction observed for both active and sham taping applications (*Robinson et al., 2019*). This implies that the effectiveness of DT in addressing disability is context-dependent, and its impact varies across different musculoskeletal conditions.

### Effect on well-being (WHO-5)

The WHO-5 Index, a measure of psychological well-being, exhibits a progressive increase from baseline (53.4) to post-intervention (57.6) and extends to follow-up (65.8). The improvement percentage indicates a substantial 23.2% positive change in DTC and 19.9% in STC group from baseline to follow-up. This finding suggests that the intervention involving DT positively influenced the mental and emotional states of the participants, implying that DT, as part of a comprehensive treatment approach, may be effective in addressing psychological well-being among patients with CNSNP. Further discussions and analyses could explore the specific elements of psychological well-being that were influenced and the potential mechanisms through which DT may have contributed to this improvement. Additionally, considering the broader clinical implications of these findings could be beneficial for healthcare practitioners and researchers working in the field.

### Immediate and long term effect of DT

The body of literature on taping techniques has significantly expanded in recent years (*Thelen, Dauber & Stoneman, 2008*). Studies tend to emphasize immediate gains (*Tudini et al., 2023*) more prominently than short or long-term effects of tape application. However, it is noteworthy that the majority of research is focused on KT rather than DT. Our study showed a statistically significant short-term effect of DT after a 4-week intervention period. Likewise, a single-blind randomized clinical trial involving 30 patients with myofascial pain syndrome reported significant improvements in pain, range of motion (except for neck extension), and neck disability in the treatment group compared to the control group, both in the short and long term with KT (*Rasti & Shamsoddini, 2018*).

## STRENGTHS AND LIMITATIONS

The acknowledgment of limitations in this trial is essential for informing the design of future studies and exercising caution in clinical decision-making regarding the application

of taping. The study design was not optimal for measuring the placebo effect of sham taping. Given the lack of clear biomechanical reasoning behind the taping effects, relying on self-reported outcome measures in this trial may not guarantee the biological impact of DT in the study population. Some studies have reported substantial immediate beneficial effects of taping in improving pain and functions, which may fade after the taping period. However, this study did not measure the immediate effects of taping for the reported outcome measures. While the DTC group participants were taped by two trained physiotherapists, this study did not account for potential physiotherapists, the differences or consistency in the force applied during taping between providers and sessions. Nevertheless, as an adequately powered trial and one of the few prospectively registered trials in the region, the findings indicate the potential clinical application of DT among CNSNP patients in improving self-reported pain, disability, and well-being. It is crucial to note that the level of physical activity was not measured using a validated instrument. Future studies are warranted to explore the objective benefits of dynamic taping in painful and disabling conditions where taping is indicated.

## CONCLUSION

The addition of DT to the standard physiotherapy care revealed improvements in pain, functional disability, and well-being in patients with CNSNP compared to the sham group. Dynamic taping is a non-invasive, painless procedure with minimal side effects and is well-tolerated. However, the clinical significance, especially beyond placebo effects, in painful musculoskeletal conditions requires further validation. Considering the nature of this study and the acknowledged limitations, caution in interpreting the findings is warranted.

### Funding

This study is supported *via* funding from Prince Sattam bin Abdulaziz University project number (PSAU/2023/R/1444) Saudi Arabia. The funders had no role in study design, data collection and analysis, decision to publish, or preparation of the manuscript.

### Grant Disclosures

The following grant information was disclosed by the authors:
Prince Sattam bin Abdulaziz University: (PSAU/2023/R/1444).

### Competing Interests

Faizan Zaffar Kashoo is an Academic Editor for PeerJ.

### Author Contributions

- Mohammad Sidiq conceived and designed the experiments, performed the experiments, analyzed the data, prepared figures and/or tables, authored or reviewed drafts of the article, and approved the final draft.

- Aksh Chahal conceived and designed the experiments, performed the experiments, analyzed the data, prepared figures and/or tables, authored or reviewed drafts of the article, and approved the final draft.
- Balamurugan Janakiraman conceived and designed the experiments, performed the experiments, analyzed the data, prepared figures and/or tables, authored or reviewed drafts of the article, and approved the final draft.
- Faizan Kashoo conceived and designed the experiments, performed the experiments, analyzed the data, prepared figures and/or tables, authored or reviewed drafts of the article, and approved the final draft.
- Sharad Kumar Kedia conceived and designed the experiments, performed the experiments, analyzed the data, prepared figures and/or tables, authored or reviewed drafts of the article, and approved the final draft.
- Neha Kashyap conceived and designed the experiments, performed the experiments, analyzed the data, prepared figures and/or tables, authored or reviewed drafts of the article, and approved the final draft.
- Richa Hirendra Rai conceived and designed the experiments, performed the experiments, analyzed the data, prepared figures and/or tables, authored or reviewed drafts of the article, and approved the final draft.
- Neha Vyas conceived and designed the experiments, performed the experiments, analyzed the data, prepared figures and/or tables, authored or reviewed drafts of the article, and approved the final draft.
- T.S. Veeragoudhaman conceived and designed the experiments, performed the experiments, analyzed the data, prepared figures and/or tables, authored or reviewed drafts of the article, and approved the final draft.
- Krishna Reddy Vajrala conceived and designed the experiments, performed the experiments, analyzed the data, prepared figures and/or tables, authored or reviewed drafts of the article, and approved the final draft.
- Megha Yadav conceived and designed the experiments, performed the experiments, analyzed the data, prepared figures and/or tables, authored or reviewed drafts of the article, and approved the final draft.
- Shahiduz Zafar conceived and designed the experiments, performed the experiments, analyzed the data, prepared figures and/or tables, authored or reviewed drafts of the article, and approved the final draft.
- Sanghamitra Jena conceived and designed the experiments, performed the experiments, analyzed the data, prepared figures and/or tables, authored or reviewed drafts of the article, and approved the final draft.
- Monika Sharma conceived and designed the experiments, performed the experiments, analyzed the data, prepared figures and/or tables, authored or reviewed drafts of the article, and approved the final draft.
- Shashank Baranwal conceived and designed the experiments, performed the experiments, analyzed the data, prepared figures and/or tables, authored or reviewed drafts of the article, and approved the final draft.

- Mshari Alghadier conceived and designed the experiments, performed the experiments, analyzed the data, prepared figures and/or tables, authored or reviewed drafts of the article, and approved the final draft.
- Abdullah Alhusayni conceived and designed the experiments, performed the experiments, analyzed the data, prepared figures and/or tables, authored or reviewed drafts of the article, and approved the final draft.
- Abdullah Alzahrani conceived and designed the experiments, performed the experiments, analyzed the data, prepared figures and/or tables, authored or reviewed drafts of the article, and approved the final draft.
- Vijay Selvan Natarajan conceived and designed the experiments, performed the experiments, analyzed the data, prepared figures and/or tables, authored or reviewed drafts of the article, and approved the final draft.

**Human Ethics**

The following information was supplied relating to ethical approvals (*i.e.*, approving body and any reference numbers):

"The Departmental Ethics Committee, Physiotherapy Department, NIMS University has granted ethical approval to conduct the study within its facilities, affirming adherence to ethical standards and oversight for the research".

**Clinical Trial Ethics**

The following information was supplied relating to ethical approvals (*i.e.*, approving body and any reference numbers):

Departmental Ethics Committee, NIMS University Jaipur Rajasthan India.

**Ethics**

The following information was supplied relating to ethical approvals (*i.e.*, approving body and any reference numbers):

The NIMS University Departmental Ethics Committee of Physiotherapy Department issued granted Ethical Approval nu/ncpt/june/15.

**Data Availability**

The data is available in the Supplemental File.

The code book for the dataset is available at figshare: Sidiq, Mohammad (2023). Code Book Dynamic Tape. figshare. Dataset. https://doi.org/10.6084/m9.figshare.24164967.v1.

**Clinical Trial Registration**

The following information was supplied regarding Clinical Trial registration:

CTRI/2022/07/043700.

**Supplemental Information**

Supplemental information for this article can be found online at http://dx.doi.org/10.7717/peerj.16799#supplemental-information.

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
