# Peer review of "Effect of dynamic taping on neck pain, disability, and quality of life in patients with chronic non-specific neck pain: a randomized sham-control trial"

_PeerJ, doi:10.7717/peerj.16799_

## Round 0.1 · original submission · Major Revisions

One reviewer recommended rejection and 1 minor revisions, I think it is publishable if the comments are thoroughly addressed. As it was a clinical study it can never be perfect; however, it needs to be tightened up. The results are overstated and the results should be toned down if that makes sense. I would also reference this recent paper and use it as an example. Tudini F, Levine D, Healy M, Jordon M, Chui K. Evaluating the effects of two different kinesiology taping techniques on shoulder pain and function in patients with hypermobile Ehlers-Danlos syndrome. Front Pain Res (Lausanne). 2023;4:1089748. Published 2023 Jan 16. doi:10.3389/fpain.2023.1089748

With this many authors, I would verify they all qualify based on the guidelines:

Authorship Criteria
Authorship credit should be based on 1) substantial contributions to conception and design, acquisition of data, or analysis and interpretation of data; 2) drafting the article or revising it critically for important intellectual content; and 3) final approval of the version to be published. Authors should meet conditions 1, 2, and 3.

When a large, multicenter group has conducted the work, the group should identify the individuals who accept direct responsibility for the manuscript (3). These individuals should fully meet the criteria for authorship/contributorship defined above, and editors will ask these individuals to complete journal-specific author and conflict-of-interest disclosure forms. When submitting a manuscript authored by a group, the submission admin should clearly indicate the preferred citation and identify all individual authors as well as the group name. Journals generally list other members of the group in the Acknowledgments. The NLM indexes the group name and the names of individuals the group has identified as being directly responsible for the manuscript; it also lists the names of collaborators if they are listed in Acknowledgments.

Acquisition of funding, collection of data, or general supervision of the research group alone does not constitute authorship.

All persons designated as authors should qualify for authorship, and all those who qualify should be listed.

Each author should have participated sufficiently in the work to take public responsibility for appropriate portions of the content.

·

Basic reporting

The English language should be improved to ensure that an international audience can clearly understand your text. Some examples include lines 30 - 31, 83 - 84, and 96 - 97. Abbreviations should be written first with the abbreviation in parentheses before being used in the paper. Examples include: PMR, OPD, and NIMS in lines 129 - 130 among others.

The results are related to the purpose statement however may not be accurately interpreted. This will be discussed under the validity of findings.

Experimental design

The methods are not described with sufficient detail in the following areas:

1. The main concern is lack of what standard physiotherapy care entails
2. There are inconsistencies in the follow up period. Most of the manuscript states baseline, 4 weeks after, and another 2 weeks after. However in line 210 it states 4 weeks follow up but there are other instances.
3. The methods section states that a partial eta squared test will be used for effect sizes but in actuality a Cohen's D was used
4. There is no reporting in the methods of related to how 140 people with chronic non specific neck pain were recruited over the course of a few months from one center. There are also gaps related to how is it known that the taping and the therapy were consistent. Were all the patients treated by the same person? Were all the patients taped by the same person? Where and how was the tape acquired to tape 140 people 3 x per week x 4 weeks? Who trained the people who did the taping?
5. Also in the methods it Staes that it took 50 - 60 minutes to collect the outcome data which consisted of only a VAS a 10 question NDI, and a 5 question WHO survey. This should have only taken a matter of minutes. What was the way this was carried out and how was the data collected?
6. How was level of activity determined?
7. There is inconsistent language used between sham and control and dynamic. I would recommend sham as there was not a true control group that did not receive tape.
This would be not be reproducible.

Validity of the findings

This study would not be reproducible, mainly because of the lack of what standard physiotherapy is, but also for the reasons outlined above under experimental design.

There are several questions related to the results. It is not believable that of 140 people taped 3 x per week x 4 weeks that no one was lost to follow and not one person missed even one session of therapy or taping. How was this factored into the study? Was an intention to treat analysis performed? Were medical changes or any changes in health status checked over the course of the 6 weeks which could have affected the outcomes? Furthermore, not one person had an adverse reaction to the tape after being continuously taped for 4 weeks is equally not believable.

The authors note that there were significant differences in gender, occupation, physical activity, and smoking but don't discuss how they accounted for this in their results.

The NDI score is based on 50 but in table 1 the NDI scores were 43.76+ 8.15 which is greater than 50. If the authors doubled the score to present the information as a percentage, this should be documented and labelled in the chart as such.

The authors state in line 249 - 250 that there was a significant reduction in NDI scores in the dynamic tape group from post intervention to follow up, but the p = 0.16.

Table 2 states that the results are within groups but the data seems to be between groups and the within group is not presented as far as I can tell.

The authors do not use MCID's for any of the outcome measures. For instance, while there was a statistical difference in VAS scores for both interventions, the sham tape group VAS decreased from 6.2 - 5.0 which does not surpass the MCID while the dynamic tape group did. Also from post intervention to follow up, neither group surpassed the MCID and this is more consistent with the literature that there may be a short term benefit from the tape but not long term: see Mata 2022.

I believe that the authors have overstated their findings: it is interesting to consider that the therapy plus sham tape resulted in a 13.4% improvement in NDI and the dynamic tape with tension resulted in over a 35% improvement. This is an extreme difference and means that the tape was more effective than therapy by far and only adjusting tape tension caused this change. This is contrary to articles such as de Jesus 2017 that showed no difference in muscle function with different tape tensions. Silva in 2021 showed no different in function with a hip application of dynamic tape and Alahmari in 2020 found only improvement in back endurance with tape and not function. Parreira 2014 when analyzing KT tape found little evidence supporting its use. With that being said, I think that dynamic tape may be different but it is not believable that tape with tension resulted in such a dramatic change and that this change lasted 2 weeks after the study stopped. This would have to be explained more as almost all current literature supports more active treatment and this study claims that the application of tape alone results in large improvements in pain and function ( An F value of well over 200 was reported).

In the discussion the authors state that the dynamic taping was successful due to improved position sense, offloading tissues, reducing muscle fatigue, controlled deceleration loading, facilitating target muscles, and increased afferent input to the CNS, but none of these variables were measured in any way. It would be appropriate to say it is hypothesized that the taping may have had positive effects for the following reasons:

Additional comments

`I think the premise of the paper and purpose are achievable; however, the paper is inconsistent. There are large gaps rendering reproducibility impossible and the authors do not follow their own methods when reporting the results. The findings are over-stated and frankly not believable and contrary to much existing quality literature in this area. If the authors were to attempt this study again, I would recommend much closer monitoring and reporting.

Reviewer 2 ·

Basic reporting

Usually the p-value is a number. However, in the main text, the testing p-value are alway presented as a range (e.g. p<0.078). Please provide the exact p-value calculated from the statistical testing.

Experimental design

1. The author should explicitly specify the factors considered in the ANOVA model, such as the treatment group and the interaction between the treatment group and time. It would enhance clarity if the ANOVA model is presented in formula format.
2. Table 1 indicates imbalances in baseline characteristics (e.g., gender, smoking history) between DTC and STC. Therefore, it is recommended to run the ANOVA model with all unbalanced baseline characteristics included as controlling variables.

Validity of the findings

no comment

Additional comments

no comment

---

## Round 0.2 · accepted · Accept

The revision has satisfied the issues with the previous manuscript. Thank you for the attention to detail and a well-done revision.